# The intervening domain is required for DNA-binding and functional identity of plant MADS transcription factors

Xuelei Lai[1,8], Rosario Vega-Léon [2,8], Veronique Hugouvieux [1,8✉], Romain Blanc-Mathieu [1], Froukje van der Wal[3], Jérémy Lucas[1], Catarina S. Silva[1,4], Agnès Jourdain[1], Jose M. Muino [5], Max H. Nanao[6], Richard Immink [3,7], Kerstin Kaufmann [2], François Parcy [1], Cezary Smaczniak [2✉] & Chloe Zubieta [1✉]

The MADS transcription factors (TF) are an ancient eukaryotic protein family. In plants, the family is divided into two main lineages. Here, we demonstrate that DNA binding in both lineages absolutely requires a short amino acid sequence C-terminal to the MADS domain (M domain) called the Intervening domain (I domain) that was previously defined only in type II lineage MADS. Structural elucidation of the MI domains from the floral regulator, SEPALLATA3 (SEP3), shows a conserved fold with the I domain acting to stabilise the M domain. Using the floral organ identity MADS TFs, SEP3, APETALA1 (AP1) and AGAMOUS (AG), domain swapping demonstrate that the I domain alters genome-wide DNA-binding specificity and dimerisation specificity. Introducing AG carrying the I domain of AP1 in the Arabidopsis *ap1* mutant resulted in strong complementation and restoration of first and second whorl organs. Taken together, these data demonstrate that the I domain acts as an integral part of the DNA-binding domain and significantly contributes to the functional identity of the MADS TF.

[1] Laboratoire Physiologie Cellulaire et Végétale, Université Grenoble Alpes, CNRS, CEA, INRAE, IRIG-DBSCI-LPCV, Grenoble, France. [2] Plant Cell and Molecular Biology, Institute of Biology, Humboldt-Universität zu Berlin, Berlin, Germany. [3] Bioscience, Wageningen Plant Research, Wageningen University and Research, Wageningen, The Netherlands. [4] European Molecular Biology Laboratory, Grenoble, France. [5] Systems Biology of Gene Regulation, Institute of Biology, Humboldt-Universität zu Berlin, Berlin, Germany. [6] European Synchrotron Radiation Facility, Structural Biology Group, Grenoble, France. [7] Laboratory of Molecular Biology, Wageningen University and Research, Wageningen, The Netherlands. [8] These authors contributed equally: Xuelei Lai, Rosario Vega-Léon, Veronique Hugouvieux. ✉email: veronique.hugouvieux@cea.fr; cezary.smaczniak@hu-berlin.de; chloe.zubieta@cea.fr

The MADS-box genes, named after founding members *MINICHROMOSOME MAINTENANCE1* (*MCM1*, *Saccharomyces cerevisiae*), *AGAMOUS* (*AG*, *Arabidopsis thaliana*), *DEFICIENS* (*DEF*, *Antirrhinum majus*), and *SERUM RESPONSE FACTOR* (*SRF*, *Homo sapiens*), are an ancient gene family present before the ancestral split of animals, plants and fungi[1–4]. Prior to the divergence of eukaryotes into different kingdoms of life, the MADS gene family underwent a duplication event giving rise to two main lineages, the *SRF*-like and *MEF2*-like lineages, which correspond to the type I and II MADS genes in plants, respectively[5,6]. While these two MADS lineages both encode a highly conserved ~60 amino acid MADS-box DNA-binding domain (DBD) that recognises a CArG-box motif (CC-"Adenine rich"-GG), DNA-binding preferences are slightly altered due to amino acid changes in the MADS-box domain specific to each lineage[7–10]. In addition to changes in amino acids important for direct base readout, the region C-terminally adjacent to the MADS-box domain varies based on sequence alignments and available structural data for mammalian and fungal SRF-like and MEF2-like MADS transcription factors (TFs). These carboxyl-terminal sequences have been shown to be important for dimerisation and DNA binding although they do not directly contact the DNA[4,7,11]. While the MADS genes are ubiquitous in extant eukaryotes, they have undergone a significant expansion in plants, with angiosperms possessing tens of type I SRF-like and type II MEF2-like MADS genes that fulfil diverse physiological roles.

The plant type I MADS genes generally consist of one or two exons and encode TFs comprised of the MADS-box DBD and a variable C-terminal domain. The type I genes are further subdivided into three subfamilies, Mα, Mβ and Mγ, with preferential interactions between subfamilies forming heterodimeric complexes[12,13]. While generally expressed at very low levels in a tissue-specific manner, the type I genes have been shown to be important in plant reproduction and speciation, with crucial roles in female gametophyte, embryo and endosperm development[14,15]. In contrast to the simple one or two exon composition and protein structure of the type I MADS, the type II genes consist of 5–8 exons and encode TFs with a modular four-domain structure, called "MIKC"[16–18]. The MIKC domains refer to M, for MADS DBD, I for the dimerisation specifying Intervening domain, K for the coiled-coil keratin-like domain involved in dimer and tetramer formation and C for the variable C-terminal domain important for transactivation and higher-order complex formation. The type II MADS are further subdivided into the MIKC* and MIKC^C subfamilies based on the structure of the I domain, with dimerisation mainly occurring within the same subfamily, but not between subfamilies[19–21]. While all MADS TFs are believed to form dimeric complexes in order to bind DNA, the type II MADS TFs are able to tetramerise via the K domain, greatly expanding the number of possible MADS complexes that can be formed[22–24]. The diversity of heteromeric complexes the type II MADS form is hypothesised to be directly tied to their functional diversity[25]. The type II MADS play well established roles in many developmental processes including meristem identity, flowering time, fruit and seed development and floral organ identity[26].

The MADS TFs involved in floral organ specification are arguably the most well-studied MADS family members and provide an ideal model to examine function in planta. Elegant in vitro and in vivo experiments dating back decades have tried to address the question of how plant MADS TFs achieve their DNA-binding specificity and functional diversity in the context of flower and floral organ development[27,28]. Four classes of MIKC MADS TFs, class A, B, C and E, are necessary for floral organ identity. The A + E class specify sepals, the A + B + E petals, A + C stamen and C + E carpels. In *Arabidopsis*, these correspond to APETALA1 (AP1, A class), APETALA3 (AP3, B class) and PISTILLATA (PI, B class), AG (C class) and the four SEPAL-LATAs (SEP1 -4, E class). The generation of chimeric MADS TFs with swapped M domains from AP1, AP3, PI and AG demonstrated that the MADS domains of these proteins were interchangeable with respect to their in vivo function[28,29]. Even swapping in portions of human or yeast M domains, which did alter in vitro DNA binding, did not alter the function of the MADS proteins in vivo, suggesting that the physiological function of these proteins seems to be independent of the DBD[29]. Thus, tetramerisation in MIKC MADS TFs came under scrutiny as a potential determinant of DNA-binding syntax by selecting for two-site DNA binding constrained by specific intersite distances[30,31]. More recent studies of a tetramerisation mutant of SEP3 revealed that while tetramerisation contributed to in vivo function, in vitro changes in DNA-binding specificity genome-wide were more limited, putatively affecting a relatively small number of key gene targets whose regulation depends on specific spacings between CArG-box binding sites[31,32]. Thus, the fundamental question of how plant MADS TFs are able to recognise different DNA sequences critical for the regulation of different target genes, with apparently interchangeable DBDs, is still not fully resolved.

In this work, we performed structural, biochemical, genome-wide binding and in vivo studies focusing on the role of the Intervening domain. We demonstrate that an I-like domain is present in both type I and type II plant MADS TFs and that this region is required for DNA binding, affects DNA-binding specificity and alters dimerisation specificity in vitro and in yeast assays. Seq-DAP-seq experiments comparing SEP3-AG and SEP3-AG^IAP1, a chimera with the I domain from AG replaced by the I domain of AP1, revealed unique binding sites for each complex, changes in preferred site spacing and a loss of carpel and fourth whorl-specific targets for the chimeric protein. Using these same floral organ identity MADS TFs to probe in vivo function of the I domain, different chimeric constructs were introduced into the *ap1* background. Interestingly, replacing the I domain of *AG* with that of *AP1* was sufficient to confer the majority of *AP1* functions in planta including sepal and petal identity in the first and second whorls. Taken together, these data illustrate the importance and multiple roles of the I domain in DNA-binding and in planta function.

## Results

**Structure of the SEPALLATA3 MI domain.** The MADS-box and I domain of SEP3 (SEP3^MI, residues 1–90) was used in structural studies as this construct has been shown to be dimeric in solution and to bind DNA in a sequence specific manner[33]. SEP3^MI crystallised in space group C222₁ with four monomers per asymmetric unit and diffraction to 2.1 Å (Table 1). The first 18 amino acids, which are predicted to have no defined secondary structure, were disordered as were the 17 C-terminal amino acids, and no electron density was interpretable for these portions of the protein in any of the molecules. Each monomer adopts the same secondary and tertiary structure with an N-terminal alpha helix (α1; aa 18–40) and two antiparallel beta strands (β1; aa 44–48 and β2; aa 55–58). These elements make up the core MADS DBD and are conserved in the MEF2 and SRF-like MADS TFs (Fig. 1a, b). C-terminal to the beta strands is a short loop followed by an alpha helix which is contributed by the I domain (α2; aa 63–73). The orientation of the C-terminal alpha helix differs between MEF2 (PDB codes 1EGW, 1C7U, 1TQE, 3KOV) and SRF-like MADS TFs (PDB code 1HBX), with SEP3^MI resembling the MEF2 structure. Overall, SEP3^MI adopts the MADS/MEF2 fold, as predicted for type II plant MADS TFs.

**Table 1 Data collection and refinement statistics.**

|  | SEP3^MI |
|---|---|
| *Data collection* |  |
| Space group | C222₁ |
| Cell dimensions |  |
| *a*, *b*, *c* (Å) | 67.4, 67.4, 122.6 |
| α, β, γ (°) | 90, 90, 90 |
| Resolution (Å) | 48.-2.10 (2.16–2.10)ᵃ |
| $R_{sym}$ or $R_{merge}$ | 8.4 (130) |
| $CC_{1/2}$ | 99.8 (56.4) |
| $I/\sigma I$ | 15.2 (1.49) |
| Completeness (%) | 99.0 (95.6) |
| Redundancy | 6.0 (4.1) |
| *Refinement* |  |
| Resolution (Å) | 47.7-2.10 |
| No. reflections | 16519 |
| $R_{work}/R_{free}$ (%) | 20.4/23.6 (32.8/34.2)ᵃ |
| No. of atoms | 1880 |
| Protein | 1814 |
| Ligand/Ion | 0 |
| Water | 66 |
| *B factors* |  |
| Protein | 55.5 |
| Water | 53.6 |
| R.m.s. deviations |  |
| Bond lengths (Å) | 0.008 |
| Bond angles (°) | 1.24 |

ᵃHighest resolution shell.

Each SEP3^MI monomer dimerises via extensive contacts between the pairs of alpha helices and beta strands from each partner, with dimerisation required for DNA binding. In the structure, all dimers were formed by crystallographic symmetry, with dimerisation burying over 25% of the total surface area of the molecule (Fig. 1)[34]. The dimer interface includes hydrophobic interactions between residues in the N-terminal DNA-binding alpha helices, Leu28 and Leu35, and pairs of salt bridges between Glu34 and Arg24 (Fig. 1d). The formation of a four-stranded antiparallel beta sheet further stabilises the quaternary structure of SEP3 with interactions bridging the N-terminal DNA-binding helices and the C-terminal I domain helices. The I domain helices sandwich the beta sheet and lie perpendicular to the DNA-binding helices of the MADS-box domain. The I domain is anchored to the beta sheet via hydrogen bonding interactions between Glu56 and Tyr70 and pi–cation interactions between Phe48 and Arg69 of the partner monomers. Each I domain helix also interacts with its dimer partner helix via hydrophobic interactions mediated by Leu67, Tyr70 and Met63 (Fig. 1c). These residues are highly conserved in all MIKC MADS TFs from *Arabidopsis*, suggesting that the interactions are also conserved and likely important for structural stability of plant MADS TFs. Mutations R69L (present in APETALA3), R69P (present in FLOWERING LOCUS M-delta) and Y70E (present in FLM-delta) destabilised SEP3^MI based on thermal shift assays, as predicted from the structural data (Supplementary Fig. 1).

SEP3^MI was crystallised without DNA, however, multiple DNA-bound structures of the closely related MEF2 protein are available for comparison. Structural alignment of SEP3^MI with the corresponding MEF2 structure reveals a RMSD of 0.942 Å², underscoring the conservation in the MADS/MEF2 domains with the MADS/I domains of SEP3^MI (Fig. 1a, b). The residues directly contacting the DNA are contributed from the flexible N-terminus and the N-terminal helix of the MADS-box domain, with no direct contacts from the MEF2/I domain (Fig. 1e). These DNA-binding residues are highly conserved in SEP3 even though MEF2 recognises a YTA(A/T)₄TAR sequence (Y = pyrimidine, R = purine), whereas SEP3 recognises a classic CArG-box motif (CC(A/T)₆GG). The differences in DNA-binding specificity of the MEF2-like MADS TFs coupled with the high conservation of sequence and structure of the MADS-box domain itself suggest allosteric contributions from the MEF2/I domain may play a role in DNA binding. Indeed, examining the DNA binding of the I domain mutants R69L, R69P and Y70E demonstrated that these single point mutations were sufficient to abrogate the DNA-binding activity of the full-length SEP3 based on electrophoretic mobility shift assays (EMSAs) (Supplementary Fig. 1).

**Type I and II MADS possess I-like domain sequences required for DNA binding.** Structural and mutagenesis data suggest that the MEF2/I domain is critical for stability and DNA binding of type II MADS TFs. The type I plant MADS TFs do not possess an I or MEF2 domain and are more closely related to the SRF-type MADS TFs. However, the structure of SRF (PDB 1HBX) reveals a similar alpha helical region C-terminal to the MADS-box domain, albeit with a different orientation of helices from the MEF2/I domain (Fig. 1a). Protein sequence alignment of the 106 MADS TFs from *Arabidopsis* reveals that 104 of the 106 sequences possess ~15 amino acids C-terminal to the MADS domain that are structurally conserved and predicted to form alpha helices with similar physicochemical properties (Fig. 2a). In particular, positions including Arg69 and Tyr70 in SEP3 that are important for I domain anchoring to the M domain are conserved in both type I and type II MADS, while a high degree of sequence variation in the non-alpha helical region is observed (Fig. 2a and Supplementary Fig. 2). This suggests that these ~15 amino acids in type I MADS TFs may act as an I-like alpha helical domain and stabilise the structure of the DBD and influence DNA-binding specificity or both.

To further determine the role of the alpha helical region adjacent to the M domain in both type I and II MADS (called for simplicity the I region hereafter), we performed pull-down experiments and EMSAs for representative MADS TFs from *Arabidopsis*. We chose AGAMOUS-LIKE 61 (AGL61) from subclass Mα and AGAMOUS-LIKE 80 (AGL80) and PHERES1 (PHE1) from subclass Mγ for type I MADS TFs, as interaction patterns of these have been previously studied[35,36]. Pull-down experiments revealed that the M + I region of AGL61 (AGL61^MI) interacts with the M + I region of AGL80 (AGL80^MI) or PHE1 (PHE1^MI), while the M domains alone are not sufficient for interaction (Fig. 2b–e), suggesting that the I region in type I MADS TFs is required for stable dimerisation. Consistent with this, EMSAs show that while heterodimers of AGL61^MI-AGL80^MI and AGL61^MI-PHE1^MI bind to a canonical CArG-box motif, the M domains alone exhibit no DNA binding (Fig. 2f).

For the type II MADS TFs, we generated M domain and MI domain constructs for SEP3, AP1 and AG. In *Arabidopsis*, SEP3-AP1 containing MADS TF complexes define first and second whorl flower organs (sepal and petals), whereas SEP3-AG containing complexes are required for third and fourth whorl (stamen and carpel) flower organ identity. We focused on these MADS TFs as their heterocomplex formation, DNA-binding specificity and in vivo activity are well understood[10,13,22,37–41]. As expected, the MI domain of SEP3 interacts with that of AP1 and AG (Fig. 3d–f). However, in contrast to the type I MADS, the M domain of SEP3 is able to pull down the M domains of AP1 and AG (Fig. 3a, c). As AG possesses an N-terminal 16 amino acid extension, constructs with and without these amino acids were tested. Both constructs were able to interact with the M domain of SEP3 (Fig. 3a, b). Thus, unlike the type I MADS TFs tested, the M

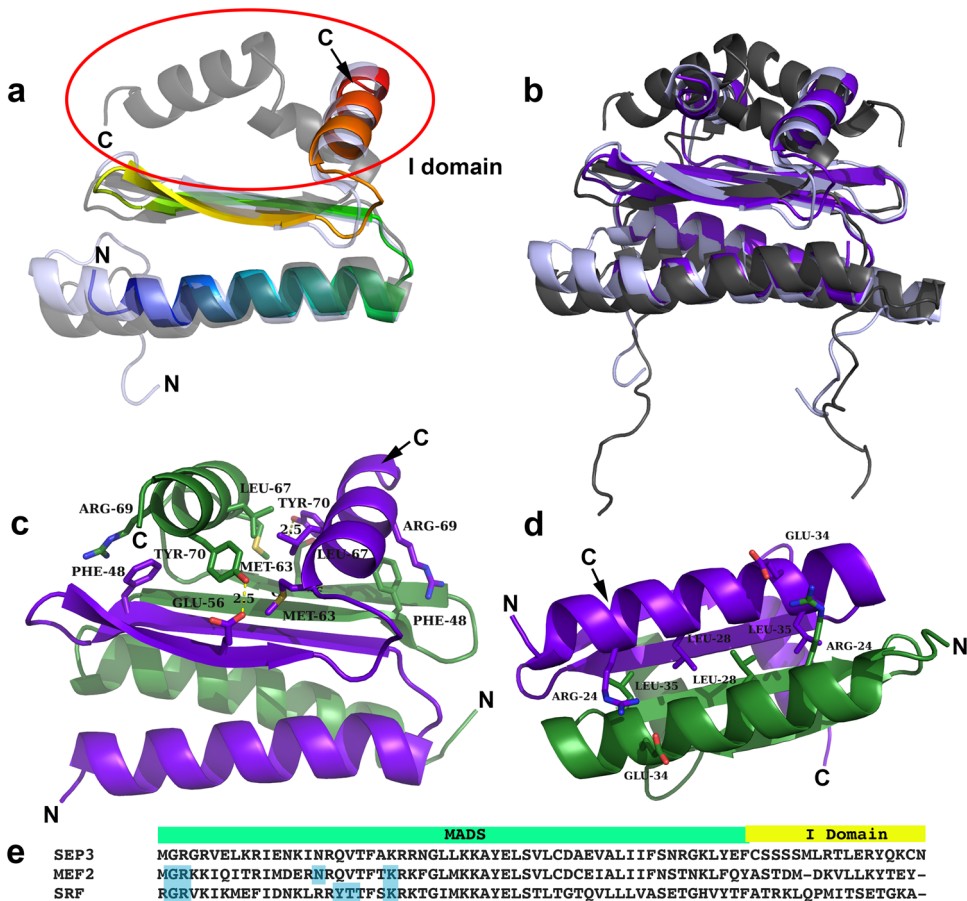

**Fig. 1 Structure and sequence of the SEP3 MI domain and corresponding SRF and MEF2 regions. a** Overlay of SEP3 dimer (rainbow) and SRF (1HBX) dimer (grey). The I region (circled in red) is alpha helical for both structures, however the contacts with the M domain differ. N and C-termini are labelled. **b** Overlay of SEP3 (dark purple) and MEF2A (3KOV, light purple) and SRF (gray) demonstrating the different conformations of the I region in the type II (MEF2-like) and I (SRF-like) MADS TFs, orientation as per **a**. **c** SEP3 dimer with one monomer in purple and one in green with N- and C-termini labelled. Amino acids important for interactions between I domains and between M and I domains are labelled and drawn as sticks. Hydrogen bonds are shown as dashed yellow lines. **d** View of the I domain with intermolecular interactions from the N-terminal alpha helices shown. **e** Partial sequence alignment of SEP3 (accession NP564214.2), MEF2 (accession AAB25838.1) and SRF (accession P11831) corresponding to the region in the crystal structure of SEP3. Residues directly contacting the DNA for MEF2 and SRF are highlighted in blue.

domain alone of the type II MADS TFs tested was sufficient for protein–protein interaction. However, the M domains alone of SEP3-AP1 and SEP3-AG were unable to bind a canonical CArG DNA motif in EMSA experiments (Fig. 3g, h), suggesting that the M domain of type II without the I domain is not sufficient to form a stable dimer for DNA binding. In contrast, homo and heterodimers of the MI domains of SEP3, SEP3-AP1, AP1, AG and SEP3-AG were all able to bind DNA under our experimental conditions (Fig. 3g, h). Taken together, these data suggest that the I domain, while possessing no direct interactions with DNA, is essential for DNA binding for both type I and type II MADS TFs, likely by stabilising the MADS TF dimer.

**Impact of the I domain on DNA-binding specificity and intersite spacing.** To further explore the role of the I domain in DNA-binding specificity, we focused on the floral organ identity-specifying MADS TFs, SEP3, AP1, AG and a chimeric version of AG, in which the AG I domain was replaced with the AP1 I domain (AG[IAP1]). Using sequential DNA-affinity purification of the full-length proteins followed by sequencing (seq-DAP-seq)[31], we determined the DNA binding of the heteromeric complexes of SEP3-AG[31] and SEP3-AG[IAP1] (this study). Unfortunately, the SEP3-AP1 heteromeric complex under the same experimental conditions did not yield data of sufficient quality for analysis. However, SEP3-AG and SEP3-AG[IAP1] strongly bound DNA and each produced highly similar replicates for data analysis (Supplementary Fig. 3). We used a highly stringent peak calling for each replicate and peaks were kept only when present in all replicates. This yielded 6347 peaks for SEP3-AG and 3552 peaks for SEP3-AG[IAP1]. These peak lists were merged and the binding intensities for the two complexes were compared for all resulting peaks. This profiling indicates that while most peaks have a similar binding intensity, several hundred exhibited at least a twofold difference in binding intensity between SEP3-AG and SEP3-AG[IAP1] (Fig. 4a, b). The same profiling between replicas for a given dataset demonstrated no such variability in binding intensity (Supplementary Fig. 3).

The top 15% sequences (1073 sequences for each complex) displaying the most extreme binding difference were used to search for specific DNA patterns that potentially code for SEP3-AG and SEP3-AG[IAP1] differential response. The TF binding sites were modelled using position weight matrices (PWM), TF flexible models (TFFM) and k-mer set memory (KSM) analysis. Due to the highly similar binding modes of the MADS family TFs, only KSM was sensitive enough to differentiate the MADS hetero-complexes (Fig. 4c, d)[42]. For the KSM, pipeline, KMAC and KSM tools from the GEM package[43] were used to search for clusters of

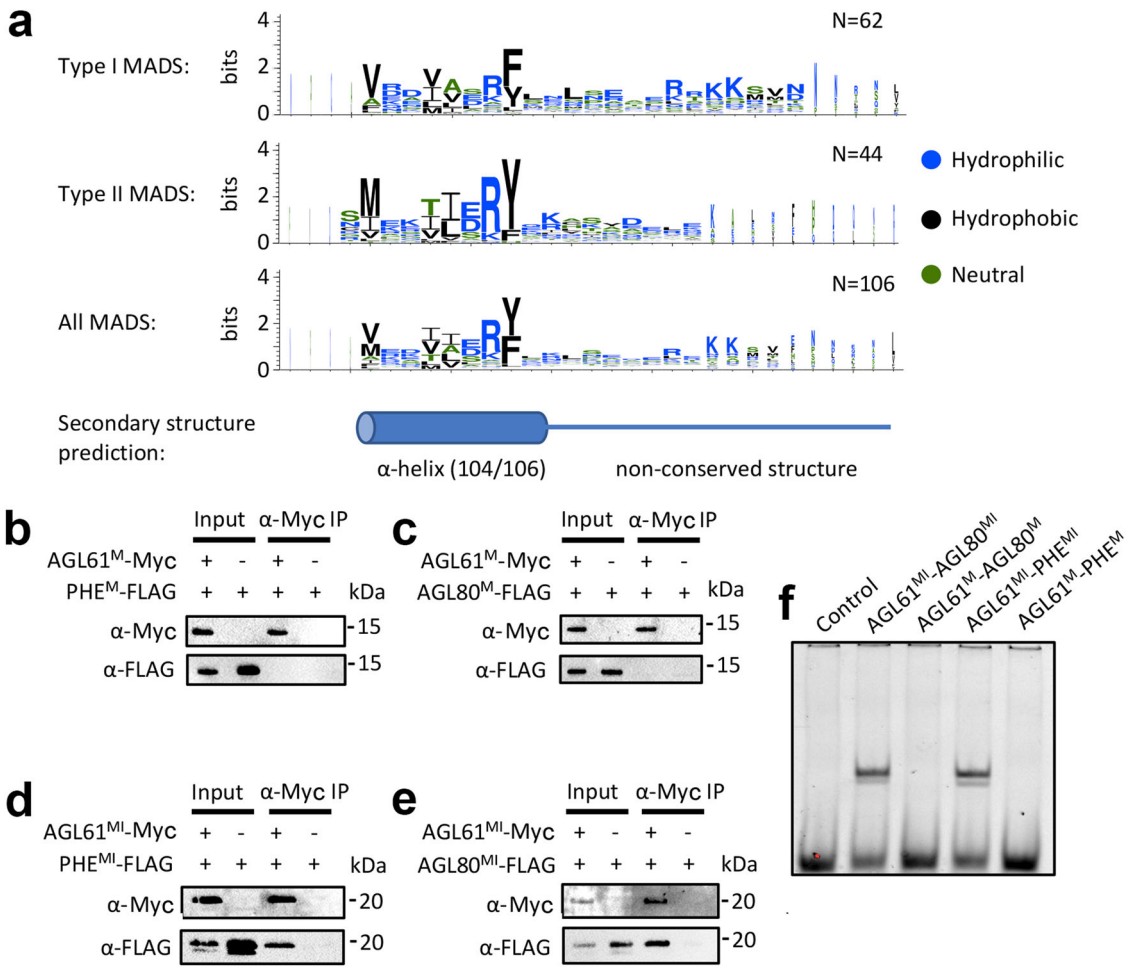

**Fig. 2 Type I MADS possesses I domain-like region which is required for both dimerisation and DNA binding. a** Amino acid enrichment of the I region (~30 amino acids C-terminal to the M domain) of type I, type II and all MADS TFs, logos generated with WebLogo[77]. The overall height of the stack in each position indicates the sequence information content at that position, while the height of the amino acid symbols within the stack indicates the relative frequency at each position. The MADS TF sequences are taken from The Arabidopsis Information Resource (www.arabidopsis.org). **b, c** Pull-down assay showing that M domain of AGL61 (AGL61$^M$) does not interact with the M domain of PHE (PHE$^M$) or AGL80 (AGL80$^M$). **d, e** Pull-down assay showing that the M domain plus the I-like region of AGL61 (AGL61$^{MI}$) interacts with the MI region of PHE (PHE$^{MI}$) and AGL80 (AGL80$^{MI}$). All assays were performed twice and a representative blot is shown. **f** EMSA assay showing that heterodimers AGL61$^{MI}$-AGL80$^{MI}$ and AGL61$^{MI}$-PHE$^{MI}$ shift a DNA sequence containing a canonical CArG-box binding site from the *SEP3* promoter, while their corresponding constructs without the I region do not exhibit any binding. All EMSAs were performed at least twice and a representative gel is shown.

short (4–20 bp) overlapping sequences (with allowances and penalties for gaps and mismatches) that are over-represented in the 600 sequences bound best by either SEP3-AG or SEP3-AG$^{IAP1}$. This analysis yielded 55 clusters for SEP3-AG and 22 clusters for SEP3-AG$^{IAP1}$ and served as k-mer-based models for prediction analysis. With AUC of 0.79 for SEP3-AG and 0.84 for SEP3-AG$^{IAP1}$, the k-mer-based models were highly predictive. Importantly, each k-mer-based model did not perform well (AUC ~0.7) in predicting bound regions from the other heterocomplex (Fig. 4d). This suggests that specific sequence patterns code for differential binding between the two heterocomplexes.

Previously, using seq-DAP-seq, we showed that SEP3-AG demonstrates preferential intersite spacing of ~47 and ~57 bp between two CArG boxes due to its cooperative binding, a contributing factor for DNA-binding specificity of MADS TF complexes[31]. Using specific bound regions of SEP3-AG$^{IAP1}$ when compared with SEP3-AG, we found that SEP3-AG$^{IAP1}$ gained new preferential intersite spacings of 25 and 34 bp and lost the SEP3-AG preferential intersite spacings (Fig. 4e). This further suggests that the I domain plays a role in DNA-binding specificity

by affecting the conformation of the tetramer and selecting for specific intersite distances.

In order to determine whether these observed in vitro differences in binding were potentially relevant in vivo, we examined published ChIP-seq datasets. Previous studies have shown that the in vitro binding of SEP3-AG relative to SEP3-AP1 in SELEX-seq correlates with the in vivo binding intensity of AG relative to AP1 in ChIP-seq for the 1500 most enriched SEP3 ChIP-seq peaks (Fig. 4f)[42]. This result suggested that specific sequence patterns detected in vitro were able to at least weakly discriminate sequences either more likely to be bound by SEP3-AG or by SEP3-AP1 in the *Arabidopsis* genome. Applying the same analysis to our seq-DAP-seq data, we found a similar weak positive correlation between the SEP3-AG binding affinity relative to that of SEP3-AG$^{IAP1}$ and the corresponding ChIP-seq derived in vivo binding, suggesting that SEP3-AG$^{IAP1}$ behaves more similarly to SEP3-AP1 than to SEP3-AG (Fig. 4f). Although modest, this correlation is supported by the observation that the genes corresponding to the 250 peaks most differentially bound by SEP3-AG relative to SEP3-AG$^{IAP1}$ exhibited a 7.6-fold

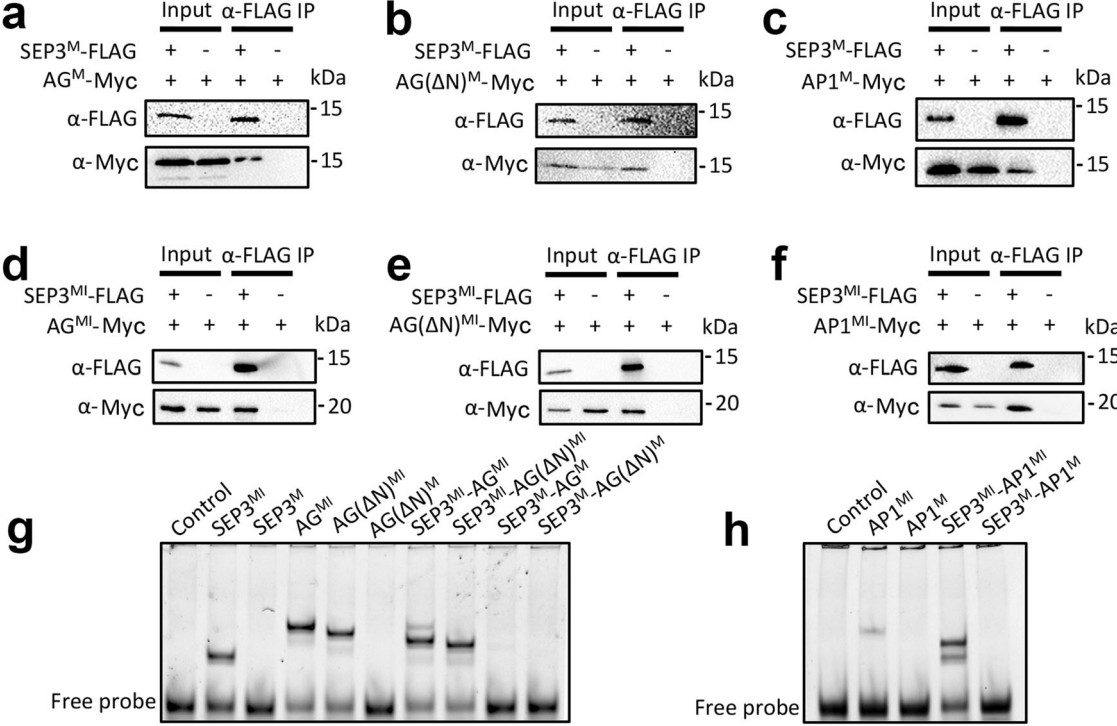

**Fig. 3 Type II MADS TFs require the I domain for DNA binding but not dimerisation. a–c** Pull-down assay showing that the M domain of SEP3 (SEP3$^M$) interacts with the M domains of AG (AG$^M$), AG with the first 16 N-terminal amino acids deleted (AG(ΔN)$^M$) and AP1 (AP1$^M$), respectively. **d–f** Pull-down assay showing that the MI domain of SEP3 (SEP3$^{MI}$) interacts with the MI domain of AG$^{MI}$, AG(ΔN)$^{MI}$ and AP1$^{MI}$, respectively. All assays were performed twice and a representative blot is shown. **g, h** EMSA assay showing that homodimers from SEP3$^{MI}$, AG$^{MI}$, AG(ΔN)$^{MI}$ and AP1$^{MI}$ and heterodimers SEP3$^{MI}$-AG$^{MI}$, SEP3$^{MI}$-AG(ΔN)$^{MI}$ and SEP3$^{MI}$-AP1$^{MI}$ shift a DNA sequence containing a canonical CArG-box binding site (as per Fig. 2), while their corresponding constructs without the I domain cannot, suggesting that the I domain in type II MADS TFs is required for DNA binding. All EMSAs were performed at least twice and a representative gel is shown.

enrichment for genes involved in "carpel development" GO term (FDR = $4.2 \times 10^{-3}$). However, no clear GO term enrichment was detected for genes corresponding to the 250 regions best bound by SEP3-AG$^{IAP1}$ relative to SEP3-AG. This may be due to the relatively small number of genes corresponding to GO terms for "petal development" (4 genes) and "sepal development" (13 genes), suggesting an incomplete list of genes for these GO terms. Taken together, the binding differences between the SEP3-AG$^{IAP1}$ and SEP3-AG complex suggest that the AG$^{IAP1}$ protein has lost some AG identity and gained AP1-like identity.

**Impact of the I domain on protein interaction specificity.** Next, we sought to understand to what extent the I domain could confer protein–protein interaction specificity as it plays an important role in dimerisation and dimer stability[27]. AP1 and AG share SEP3 as a binding partner in the molecular model predicting flower organ specification, but do exhibit differential heterodimerisation capabilities with other MIKC$^c$ MADS TFs[13,22]. In order to probe the function of the I domain in dimerisation specificity, a matrix-based yeast two-hybrid screen for AP1, AG and AG$^{IAP1}$ against all *Arabidopsis* type II MIKC$^c$ MADS TFs was performed. The targets were cloned into both bait and prey vectors and only interactors for both replica and in all three selection media were scored as positive interactions (Supplementary Fig. 4). This may underestimate the true number of binding partners, however, it reduces the number of false positives. Using this strict cutoff for protein–protein interaction events, AP1 interacted with seven different partners, of which three were specific for AP1 and four were in common with AG, including the SEP clade members SEP1 and SEP3. AG interacted with ten partners of which six were exclusive with respect to AP1

interacting partners. AG$^{IAP1}$ interacted with ten partners, including two out of the three AP1-specific interactions and lost four AG-specific partners (Fig. 4g). Interestingly, both SOC1 and AGL24, known AP1 binding partners[13] interacted with the AG$^{IAP1}$ chimera but not with AG. These data indicate that the I domain of AP1 contributes to heterodimerisation specificity of the chimera.

**In planta function of the I domain.** *Ap1* mutants (strong *ap1-7* and intermediate *ap1-11*) exhibit a non-ambiguous flower phenotype including the homeotic conversion of the first whorl sepals into bract-like organs, as characterised by their leaf-like epidermal morphology and the formation of buds in their axils and the absence of second whorl organs (Supplementary Fig. 5)[44,45]. To examine the role of the I domain in vivo, complementation of the *ap1-11* and *ap1-7* mutants with different *AG/AP1* chimeric constructs was examined. Primary transformants expressing a battery of chimeric domain swap constructs, *AG$^{MIKAP1}$*, *AG$^{MIIAP1}$*, *AG$^{MAP1}$*, *AG$^{IAP1}$*, *AG$^{KCAP1}$* and *AP1$^{IAG}$*, under the control of the *AP1* promoter, revealed a spectrum of complementation for the first and second whorls in the intermediate *ap1-11* mutant (Fig. 5).

*AG$^{MAP1}$* (Fig. 5b, panel 3) expressing plants were poorly complemented with a reduced number of first and second whorl organs and only partial organ identity recovery with petaloid and sepaloid organs. *AG$^{KCAP1}$* (Fig. 5b, panel 5) also complemented poorly, with most of the plants showing carpeloid sepals and with only partial recovery of 1–3 sepals in half of the plants. In contrast, almost complete complementation was present in plants expressing *AG$^{MIAP1}$* (Fig. 5b, panel 2), with the majority of the transformants showing four sepals or petaloid sepals in the first

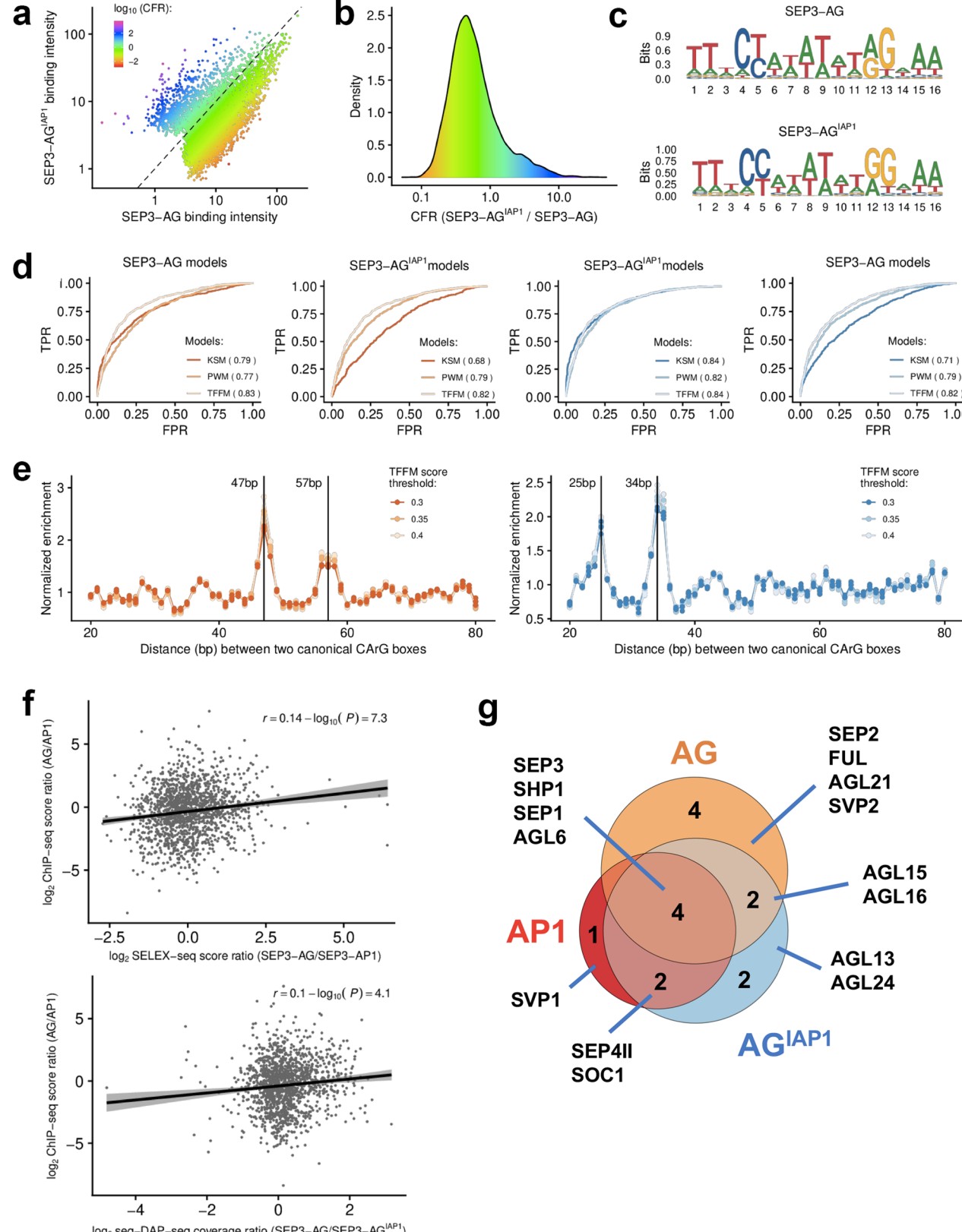

whorl and two to four petals in the second whorl. This was similar to plants expressing $AG^{MIKAP1}$ (Fig. 5b, panel 1), in which first and second whorl organ identity was restored for a majority of the plants with occasional reduced number of organs in the second whorl and lateral flower formation. *AP1* plants with the I domain of AG ($AP1^{IAG}$) (Fig. 5b, panel 6) showed a similar

phenotype as $AG^{MAP1}$ (Fig. 5b, panel 3). Strikingly, $AG^{IAP1}$ (Fig. 5b, panel 4) expressing plants exhibited four sepaloid organs and three to four petals or petaloid organs in more than half of the plants, suggesting that the I domain acts as the smallest unit conferring the most AP1 function to the $AG^{IAP1}$ chimera (Supplementary Table 2).

**Fig. 4 DNA-binding and protein interactions patterns. a** Comparison of SEP3-AG and SEP3-AGI$^{IAP1}$ seq-DAP-seq binding intensity (log$_{10}$ of reads per kb per million of reads mapped in bound regions) and colour coded by purple-blue (SEP3-AG$^{IAP1}$ specific) to orange-red (SEP3-AG specific) according to log$_{10}$ of SEP3-AG$^{IAP1}$/SEP3-AG. **b** Density plot showing data as per **a**. **c** Logos derived from PWM-based models obtained for SEP3-AG and SEP3-AG$^{IAP1}$. **d** Predictive power of TFBS models. Models are built using 600 sequences best bound by each of the two heterocomplexes and are searched against 1073 SEP3-AG (orange) and 1073 SEP3-AG$^{IAP1}$ (blue) specific regions, defined as the top 15% of sequences that are most strongly bound by one complex relative to the other. Matrix-based models (PWM and TFFM) are not able to differentiate SEP3-AG and SEP3-AG$^{IAP1}$ binding, whereas k-mer-based analysis is able to better predict binding for the respective datasets. **e** SEP3-AG favours intersite spacings of 47 and 57 bp based on SEP3-AG-specific regions. SEP3-AG$^{IAP1}$ favours intersite spacings of 25 and 34 bp based on SEP3-AG$^{IAP1I}$ specific regions. **f** Top, published SELEX-seq for SEP3-AP1 and SEP3-AG[78] comparing the normalised score ratios (SEP3-AG/SEP3-AP1) for SELEX-seq and score ratios (AG/AP1) ChIP-seq at 1500 SEP3 best bound loci in ChIP-seq show a positive correlation, suggesting that SEP3-AP1 and SEP-AG bind different sequences in vivo and that in vitro binding is able to differentiate bound sequences that are more SEP3-AP1-like versus SEP3-AG-like. Bottom, SEP3-AG and SEP3-AG$^{IAP1}$ seq-DAP-seq coverage as per SELEX-seq scores. A positive correlation is observed suggesting that, in vitro, the swap of AP1 I domain in AG is able to recover some of the binding specificity of SEP3-AP1. **g** Yeast two-hybrid assays using AG, AP1 and AG$^{IAP1}$ as bait against MIKC$^C$ MADS TFs in *Arabidopsis*. Data show that AG$^{IAP1}$ loses AG interactors and gains AP1 interactors.

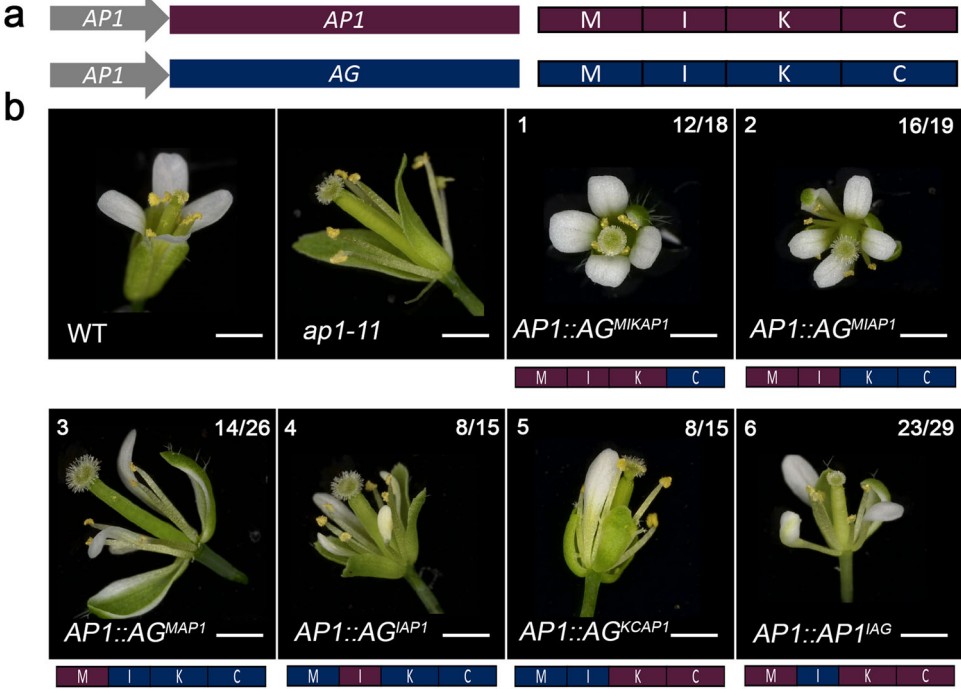

**Fig. 5 Primary transformants in the *ap1-11* mutant background exhibit a spectrum of complementation. a** Schematic of the constructs and proteins produced with AP1 protein in purple and AG in dark blue. Domains are labelled MIKC. **b** Flower phenotypes of T1 transformants ($n \geq 10$). All transformants were in the *ap1-11* background. Phenotypes for WT and *ap1-11* are shown. The domains corresponding to AP1 and AG are coloured as per **a** and shown schematically below each panel labelled 1–6 in the top left. The number of primary transformants exhibiting a given phenotype is listed in the top right of panel 1–6.

To confirm the phenotype of the *AG$^{IAP1}$* expressing plants, three independent homozygous lines were generated and examined for *AP1*, *AG* and *AG$^{IAP1}$* under the *AP1* promoter in the strong *ap1-7* mutant background (Fig. 6 and Supplementary Figs. 6–8). *AG$^{IAP1}$ ap1-7* lines produced flowers very similar to WT (Fig. 6).

In these plants, whorl 1 of each flower was made of four sepal-like organs and axillary buds were never observed as opposed to the *ap1-7* mutant (Figs. 6 and 7). In addition, between 66 and 82% of the flowers in three independent lines homozygous for the transgene showed four petal-like organs (Figs. 6 and 7). Scanning electron microscopy (SEM) observations of epidermal cells from whorls 1 and 2 of *AG$^{IAP1}$* expressing lines showed elongated cells characteristic of sepals and conical cells characteristic of petals, respectively (Fig. 7 and Supplementary Fig. 7). Detailed inspection of sepal and petal surfaces by SEM from the *AG$^{IAP1}$* expressing

plants revealed the presence of a few clusters of leaf-like and stamen identity cells in whorls 1 and 2, respectively (Fig. 7 and Supplementary Fig. 8). In these three lines, 10–25% of the flowers showed petals with only petal identity cells, while 75–90% of the flowers show at least one petal with a small cluster of stamen identity cells (Supplementary Fig. 8). This indicates incomplete complementation of *AP1* function by the chimera. However, *AP1: AG* was unable to even partially complement *AP1* function in the first and second whorl. Inspection of the cell surface in *ap1-7 AP1: AG* plants of the first whorl showed characteristic carpel cells (replum, style and papilla) and an absence of organs in the second whorl (Supplementary Fig. 6). *AG$^{IAP1}$* therefore possesses the ability to trigger AP1-specific developmental programs and has lost much of its ability to trigger *AG*-specific organ development, indicating that the I domain of AP1 plays a major role in conferring AP1 functional identity to AG$^{IAP1}$ in vivo.

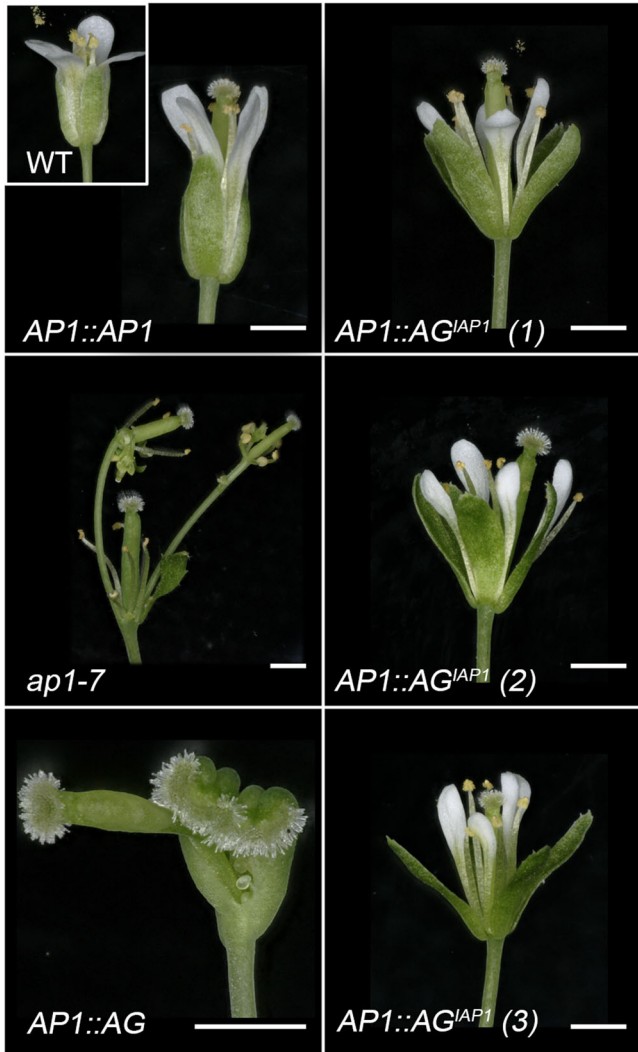

**Fig. 6 AP1:AG^IAP1 expression largely complements the *ap1*-7 flower phenotype.** *ap1*-7 lines expressing either *AP1* (*AP1::AP1*), *AG* (*AP1::AG*) or *AG^IPA1* (*AP1::AG^IAP1*) under the control of the *AP1* promoter were grown at 22 °C in long days. A typical WT flower is shown as an insert in the upper left panel. Representative flowers for three independent *AP1::AG^IAP1* lines are presented and one representative flower from *AP1::AG* and *AP1::AP1* expressing lines. While the first whorl in *ap1*-7 is replaced by bract or stipule with axillary buds and petals are missing in the second whorl, *AP1::AG^IAP1* expression restores WT first and second whorl organs, while *AG* expression triggers carpel development in the first whorl and absence of petals in the second whorl. Scales bars = 1 mm.

## Discussion

TF function depends on selective binding of specific sequences in the genome. The MADS TF family has dramatically expanded in the plant lineage over the course of evolution and fulfils many roles throughout the lifecycle of the plant. MADS constitute one of the largest plant TF families and are almost evenly divided into two types, I and II, represented structurally by SRF and MEF2. The MADS-box DBD, encoded by a single exon, is either SRF-like or MEF2-like and exhibits little sequence variation even amongst different eukaryotic kingdoms of life. How the MADS TF family is able to regulate diverse developmental processes and different target genes with such a highly conserved DBD has been a fundamental question leading to the speculation that regions distal to the DBD may play a role in DNA-binding specificity either via the formation of higher-order complexes, the recruitment of

ternary factors or via allosteric alterations in DNA-binding specificity of the MADS-box domain.

Domain swap experiments performed over 25 years ago suggested that the I domain played a key role in DNA-binding specificity and dimerisation in floral organ identity MADS TFs, although the mechanism of this was unclear[28,41,46]. Recent work examining the evolution of the class B MADS in grasses has further underscored the critical role of the I domain in homo and obligate heterodimerisation. Examination of class B MADS from different Poales taxa, mutagenesis studies and complementation experiments reveal that the I domain determines protein partner binding specificity as well as in planta functional specificity[47,48]. Changes in the genome-wide binding of the mutant proteins was not determined, although based on altered phenotypes, DNA binding would be affected. Based on the in vitro and in vivo data presented here, the I region or I domain should be considered as a fundamental component of the DNA-binding module as it is essential for DNA binding for both type I and II MADS TFs. Examination of the structure of the MI domains of SEP3 in comparison with MEF2 and SRF coupled with secondary structure predictions reveals that the I region is always alpha helical and interacts extensively with the beta sheet of the M domain, albeit with different orientations. This suggests that allosteric effects of the I region in type I and II MADS TFs, due to variability in amino acid composition, length and secondary structure orientation, will influence the MADS-domain conformation and tune DNA binding and specificity. In addition, the I region alters dimer partner specificity, altering the repertoire of heteromeric MADS complexes able to be formed and resulting in additional functional diversity in vivo.

With the introduction of the floral quartet model, the role of the I domain in determining MADS TF function specificity was supplanted by the importance of specific tetramer components triggering DNA looping as being required for gene regulation[23]. Tetramerisation allows cooperative two-site binding and looping of DNA, thus selecting for DNA-binding sites that exhibit preferred spacing[30,33,49,50]. Examination of a tetramerisation mutant of SEP3 revealed that fourth whorl carpel development and meristem determinacy require efficient tetramerisation of SEP3[32]. Recent studies comparing the genome-wide binding patterns of tetrameric versus dimeric SEP3-AG complexes have further shown that tetramerisation both increases binding affinity and plays a role in determining DNA-binding specificity via preferential binding of specific intersite distances[31]. Comparisons of genome-wide binding of SEP3-AG and SEP3-AG^IAP1, presented here, reveals that the I domain alters preferred intersite spacing between CArG-type binding sites. These data suggest a role of the I domain as a modulator of tetramerisation and highlight an additional mechanism mediated by the I domain for changing DNA-binding patterns of MADS TF complexes.

The recruitment of ternary factors, non-MADS family protein partners, has been postulated to be a major determinant of DNA-binding specificity[51]. Based on yeast two-hybrid screening, mass spectrometry and pull-down assays, additional TF partners from the NF-Y and homeodomain families as well as the orphan TF, LEAFY, have been identified for a few type II MADS TFs including OsMADS18 from rice and SHATTERPROOF, SEED-STICK, AG and SEP3 from *Arabidopsis*[24,46,52,53]. While ternary complex formation likely plays a role in differential gene regulation, the formation of such complexes has only been shown for a relatively small number of MADS TFs and how much it accounts for the diversity of function in the MADS TF family requires further studies. As shown here, swapping the I domain of the well-studied floral organ identity MADS TF, AP1 (first and second whorl organs), into the third and fourth whorl-specific TF, AG, results in strong complementation of the *ap1*

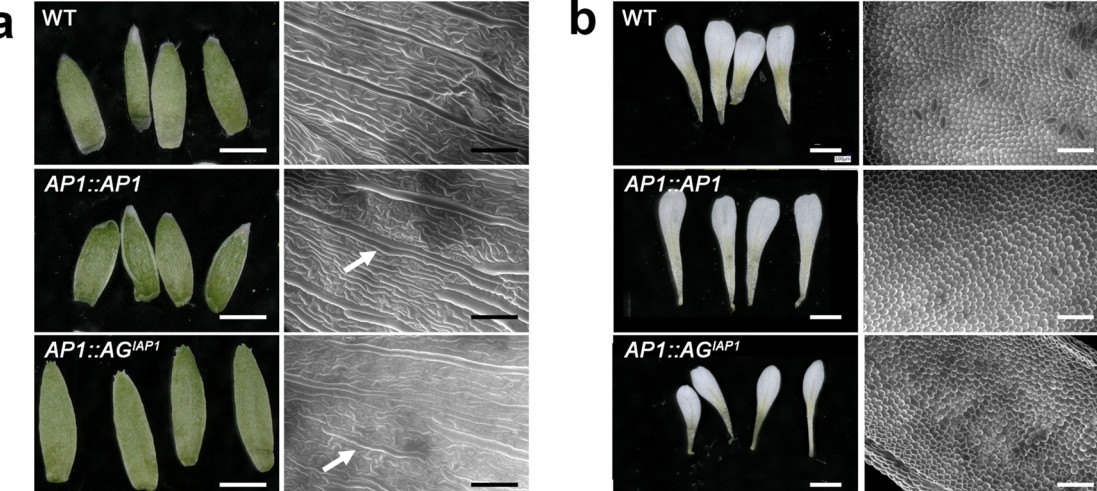

**Fig. 7 First and second whorls cell identity are complemented in *ap1-7* plants expressing AG^IAP1.** First (**a**) and second (**b**) whorl organs were removed from WT, and flowers from *AP1* or *AG^IAP1* expressing *ap1-7* plants. First whorl organs in *AG^IAP1* expressing plants are slightly longer compared to WT and *AP1* expressing plants (**a**, left panel). Second whorl organs in *AG^IAP1* expressing plants are slightly smaller compared to WT and *AP1* expressing plants (**b**, left panel). Epidermal cell identity, observed by SEM, shows characteristic WT elongated sepal cells in *AP1* and *AG^IAP1* expressing plants (**a**, right panel) and characteristic WT conical cells in petals from *AP1* and *AG^IAP1* expressing plants (**b**, right panel). A small number of epidermal cells typical of leaves are also seen in *AG^IAP1* expressing plants and to a lesser extent in *AP1* expressing plants (arrow). Scale bars indicate 100 µM for SEM images and 1 mm for organ photographs.

loss-of-function phenotype by the chimeric construct. The experiments keep the K- and C-terminal domains of AG intact and these domains have been postulated to be important for neofunctionalisation[54] and/or the recruitment of ternary factors[55]. We cannot completely exclude that ternary factors recruited via the I domain could be responsible for the *AP1*-like activity of *AG^IAP1* in planta. For example, OsMADS18 uses the MI domains to recruit OsNF-YB1 based on yeast two-hybrid assays[46]. However, it is more likely that the combination of tuning DNA-binding specificity and heterodimer partner selection by the I domain accounts for the majority of AP1 activity of the AG^IAP1 chimera.

Based on the in vitro and in vivo experiments presented here, the I region is absolutely required for DNA binding with the minimal DBD consisting of the M plus the I region, as opposed to the M domain alone. As shown here, the I domains of AP1 and AG also play an important role in DNA-binding specificity and dimerisation specificity. The I domain is able to tune the function of plant MADS TFs and, in the case of AP1, acts as a major determinant of functional identity in concert with the M domain. Further experiments will be required to determine how generalisable these results are to the wider MADS TF family (type I and II) and whether the I region broadly acts as a primary factor in MADS TF functional identity in planta.

## Methods

**SEP3 MI domain protein expression and purification.** SEP3 MI (aa 1–90)[33] was cloned into the pETM41 vector using the NcoI-NotI restriction sites to obtain His6-MBP translational protein fusion. Recombinant SEP3 MI protein was expressed in *E. coli* BL 21 Codon Plus cells. Cells were grown at 37 °C to an $OD_{600}$ of 0.8–1 after which time the temperature was reduced to 15 °C and protein expression induced by addition of 1 mM of IPTG for 12 h. Cells were harvested by centrifugation and the cell pellet resuspended in 25 mM $KH_2PO_4$ pH 7 buffer containing 10% glycerol, 500 mM NaCl, 2 mM Tris(2-carboxyethyl)phosphine (TCEP) and cOmplete protease inhibitors (Roche). Cells were lysed by sonication and cell debris pelleted at 25,000 rpm for 40 min. The soluble fraction was applied to a 1 ml Ni-NTA column and the protein eluted with resuspension buffer + 200 mM imidazole. The protein was dialysed against 25 mM Tris, pH 8.0, 300 mM NaCl and 1 mM TCEP. The protein was then applied to a heparin column to remove any bound DNA using a salt gradient from 300 mM to 2 M NaCl. The protein was dialysed overnight in the presence of TEV protease (10:1) to cleave the His-MBP tag at 4 °C. The protein was then passed over a Ni-NTA column to

deplete the His-MBP and any uncleaved protein. A second heparin column was run to remove any His-MBP and to obtain pure SEP3 MI. The protein was concentrated to ~6 mg/ml and used for crystallisation trials. SEP MI mutant constructs were generated using the QuikChange (Agilent) protocol according to the manufacturer's instructions. Primers used for mutagenesis are given in Supplementary Table 3.

**Protein crystallisation, data collection and refinement.** SEP3 MI at a concentration of 6 mg/ml was mixed at a 1:1 ratio with potassium sodium tartrate tetrahydrate (0.2 M), bis-tris propane (0.1 M, pH 7.5) and 20% PEG 3350. The protein crystallised over 20 days at 4 °C forming a diamond shaped single crystal. Fifteen percent glycerol was used as a cryoprotectant and the crystal flash frozen in liquid $N_2$. Diffraction data were collected at 100 K at the European Synchrotron Radiation Facility, Grenoble, France, on ID29 at a wavelength of 1.0 Å. Indexing was performed using MXCube[56] and the default optimised oscillation range and collection parameters used for data collection. The dataset was integrated and scaled using the programs *XDS* and *XSCALE*[57]. The structure of the MEF2 dimer (PDB code 3KOV) was used as a search model. Molecular replacement was performed using Phaser[58]. Model building was performed using Coot[59] and all refinements were carried out in Refmac[60]. The structure quality was assessed using MolProbity. Data collection and refinement statistics are given in Table 1. The structure is deposited under PDB code 7NB0.

**In vitro pull-down assays.** The following constructs from type I and type II MADS TFs are cloned into pTnT vector for in vitro pull-down assays. Type I MADS TFs include AGL61^M(63-122)-5Myc, AGL61^MI(63-155)-5MyC, PHE^M(1-60)-3FLAG, PHE^MI(1-95)-3FLAG, AGL80^M(1-61)-3FLAG and AGL80^MI(1-88)-3FLAG, and type II MADS TFs include SEP3^M(1-57)-3FLAG, SEP3^MI(1-90)-3FLAG, AG^M(1-73)-5MyC, AG^MI(1-107)-5MyC, AG(ΔN)^M(17-73)-5MyC, AG(ΔN)^MI(17-107)-5MyC, AP1^M(1-57)-5MyC and AP1^MI(1-92)-5MyC. A list of primers used for cloning is provided in Supplementary Table 4. Proteins were produced using in vitro transcription and translation system using wheat germ extract following manufactures' instructions (SP6 High Yield Expression System from Promega). Briefly, purified plasmids were used as input in a 2 h 25 °C incubation reaction. For single input reactions, 1 µg of plasmid was used in a 25 µl reaction volume. For double input reactions, 1 µg plasmid of each construct was used in a 25 µl reaction volume. Ten percent of each input reaction (i.e., 2.5 µl) was used as input for western blots. The rest of the reaction was used for pull-down experiments. Briefly, each reaction was completed to 100 µl using PBS buffer (150 mM sodium phosphate, pH 7.2 and 150 mM NaCl), and added with 10 µl of the appropriate magnetic beads (anti-Myc beads cat. 88842, Pierce or anti-FLAG beads cat. M8823, Merck Millipore) and incubated for 1 h at 4 °C. The beads plus protein solution suspension was then placed on magnets, and the supernatant was discarded. The beads were washed with PBS buffer four times, and SDS-PAGE loading dye was added to the beads and boiled for 5 min at 95 °C. Western blots were used to assess the pull-down results using anti-c-Myc (cat. R951-25 Thermo

Fisher) and anti-FLAG antibodies (cat. A8592 Sigma-Aldrich) at a dilution of 5000× for both antibodies.

**EMSA experiments**. Proteins for EMSAs experiments were produced as described above. EMSAs were performed as described with 10 nM DNA labelled with Cy5 (Eurofins) using a 103-bp DNA fragment containing two CArG boxes belonging to the SEP3 promoter[32]. For each EMSA, a negative control was run, 'free probe' in which the in vitro translation assay was done with pTNT vector without any insert and incubated with the DNA probe.

**Yeast two-hybrid screening**. AP1, AG and the chimeric construct, AG[IAP1] (residues 74–107 of AG replaced by residues 58–92 of AP1) were cloned into pENTR/D-TOPO® (Kan) gateway entry vector. Subsequently, gateway LR reactions were performed using pDEST32 (pBDGAL4) and pDEST22 (pADGAL4) destination vectors. The resulting pDEST32-AG, pDEST32-AP1 or pDEST32-AG[IAP1] expression construct was transformed into yeast strain PJ69-4α, and pDEST22-AG, pDEST22-AP1 or pDEST22-AG[IAP1] into PJ69-4A, followed by an autoactivation screen for the bait vector as described[56]. Upon confirmation that the AG[IAP1] bait did not possess any autoactivation capacity, a matrix-based Y2H screening was performed following the protocol described in ref. [61]. Bait and prey clones of all native *Arabidopsis* type II MIKC^c MADS TFs were generated previously[13] and were screened against AP1, AG and AG[IAP1]. Growth of yeast and hence protein–protein interaction events were scored after 7 days of incubation at 20 °C on selective medium. Three different selective media were used: SD lacking leucine, tryptophan and histidine (-LWH) and supplemented with 1 mM 3-amino-1,2,4-triazole (3-AT); SD-LWH + 5 mM 3-AT and SD lacking L, W and adenine. Mating was performed twice in independent experiments and only combinations of MADS TFs giving growth on all three selection media and in both replica were scored as interaction events.

**Seq-DAP-seq**. Seq-DAP-seq for SEP3-AG[IAP1]complex was performed as described previously[31]. Briefly, 2 μg of each purified plasmid (pTnT-SEP3-3FLAG and pTnT- AG[IAP1]−5MyC) was used as input in a 50 μl TnT reaction incubated at 25 °C for 2 h (Promega). The reaction solution was then combined with 50 μl IP buffer (PBS supplemented with 0.005% NP40 and proteinase inhibitors (Roche)) and mixed with 20 μl anti-FLAG magnetic beads (Merck Millipore). Following 1 h incubation at room temperature, the anti-FLAG magnetic beads were immobilised, and washed three times with 100 μl IP buffer. Protein complexes were eluted with 100 μl IP buffer supplemented with 200 μg/ml 3xFLAG peptide (Merck Millipore). The eluted protein was then immobilized on anti-c-Myc magnetic beads (Thermo Fisher) and washed three times with 100 μl IP buffer to isolate homogeneous SEP3-AG[IAP1]complex. The purified protein complex, while still bound on anti-c-Myc magnetic beads, was incubated with 50 ng DAP-seq input library pre-ligated with Illumina adaptor sequences. The reaction was incubated for 90 min, and then washed six times using 100 μl IP buffer. The bound DNA was heated to 98 °C for 10 min and eluted in 30 μl EB buffer (10 mM Tris-Cl, pH 8.5). The eluted DNA fragments were PCR amplified using Illumina TruSeq primers for 20 cycles, and purified by AMPure XP beads (Beckman). The libraries were quantified by qPCR, pooled and sequenced on Illumina HiSeq (Genewiz) with specification of paired-end sequencing of 150 cycles. Each library obtained 10–20 million reads. The seq-DAP-seq was performed in triplicate.

**Seq-DAP-seq data analysis**. Reads processing and peak calling for SEP3-AG and SEP3-AG[IAP1] were performed as previously described[31]. Briefly, reads were checked using FastQC (http://www.bioinformatics.babraham.ac.uk/projects/fastqc/) and adaptor sequences removed with NGmerge[62] and mapped with bowtie25[63] onto the TAIR10 version of the *A. thaliana* genome (www.arabidopsis.org), devoid of the mitochondrial and the chloroplast genomes. The duplicated reads were removed using the samtools rmdup program[64]. The resulting alignment files for each sample were input to MACS2[65] to call peaks using the input DNA as control. Consensus peaks between replicates were defined using MSPC[66] (P value cutoff = 10$^{-4}$) for each experiment. Each consensus peak was scanned for possible sub-peaks, split into several peaks if needed and the peak widths were then re-sized to ±200 bp at both side of the peak maximal height. For all the resulting peaks, coverage was computed as the mean of the normalised read coverage for each replicate. This normalised coverage defines the binding intensity of a hetero-complex at a bound region.

Comparison between SEP3-AG and SEP3-AG[IAP1] was performed as follows. SEP3-AG or SEP3-AG[IAP1] was merged according to the following procedure: peaks were considered common if at least 80% of two peaks overlapped with <50% of either peak non-overlapping. Peaks that did not overlap by >50% of their length were considered new peaks. These values were chosen empirically based on visual inspection of the peaks in the Integrated Genome Browser[67]. The averaged normalised coverage from each experiment, divided by the peak size, was computed for each peak. Figure 4 was computed using R (https://www.Rproject.org) and the ggplot library[68]. The coverage fold reduction (CFR) was computed as the ratio between the mean normalised coverages in SEP3-AG[IAP1] and SEP3-AG seq-DAP-seq. The top 15% sequences with extremes CFR were considered SEP3-AG[IAP1] and SEP3-AG specific and used to search for differential DNA

patterns that potentially direct TF differential binding using sequence modelling. Detection of preferred spacings between canonical CArG boxes was performed as in ref. [31].

PWM, TFFM and KSM were obtained as follows. For each experiment, PWM, TFFM and KSM models were reconstructed out of the 600 best peaks (judged according to their averaged coverage). PWM were generated by the meme-suite, using meme-chip[69] with options -meme-minw 16, -mememaxw 16, -meme-nmotifs 1 –meme-pal. TFFM were generated using the TFFM-framework package and the PWM obtained in the meme-chip output[70]. KSM is a recently developed TF binding motif representation that consists of a set of aligned k-mers that are over-represented at TF binding sites. KSM were generated using the KMAC tool[43] with options set to search k-mers, with size ranging from 4 to 20 bp in a 300 bp sliding windows, that are enriched compare to TF unbound sequences. TF binding sites were predicted by searching the PWM, TFFM and KSM models against TF bound sequences and unbound sequences. PWM and TFFM scan was performed using in house scripts. KSM were searched using the KSM tool[43]. The best TFFM/PWM scores and the sum of KSM scores (because distinct KSM can hit to the same subject sequence) obtained for each bound regions and for unbound regions are retained to assess the prediction power of each model in an area under the receiver operating characteristic curve. In this assessment, the unbound set of regions are chosen with similar GC content, size and origin (promoter, intron, exon and intergenic) than the set of bound regions[71]. The set of unbound sequences fed to KMAC to detect enriched k-mer is different than the one used to evaluate the KSM model prediction.

**Plant material and growth conditions**. All experiments were performed using *A. thaliana* Col-0 accession. Ap1-7 allele was obtained from *cal/cal ap1-7* plants kindly provided by Justin Goodrich and the *ap1-11* (N6231) mutant was ordered from the Nottingham Arabidopsis Stock Centre. Seedlings were grown in controlled growth chambers in long day conditions (16 h light/8 h dark) at 22 °C for plant transformation and phenotype analysis. WT, *ap1* mutant and complemented lines were always grown in parallel.

**Plasmid constructs for plant transformation**. Constructs used for the primary transformant phenotypic analysis were cloned into a modified version of pMDC32 binary vector[72], where the 2x35S promoter was replaced with the AP1 promoter. AG[MAP1], AG[MIAP1], AG[MIKAP1], AG[KCAP1], AG[CAP1], AP1[IAG] and AG[IAP1] coding sequences were amplified from pTNT-AG[MAP1], pTNT-AG[MIAP1], pTNT-AG[MIKAP1], pTNT-AG[KCAP1], pTNT-AG[CAP1], pTNT-AP1[IAG] and pTNT-AG[IAP] vectors and subsequently inserted by XbaI restriction enzyme downstream of the AP1 promoter (Supplementary Table 5). pTNT-CDS vectors were generated through Gibson assembly (NEB).

AP1::AP1, AP1::AG and AP1::AG[IAP1] vectors were generated using Gibson assembly in the vector backbone pFP100[73]. AP1 promoter and OCT terminator were PCR amplified from pML-BART-AP1pro:AP1-AR plasmid[74]. AP1, AG and AG[IAP1] coding sequences were amplified from pTNT-AP1, pTNT-AG and pTNT-AG[IAP1] vectors. Primer sequences are given in Supplementary Table 6.

**Plant transformation and floral phenotype analysis**. All the plant transformations were performed using *ap1* mutant plants following the floral dip method[75]. For primary resistant analysis in *ap1-11* intermediate allele, T1 seeds were sown on the 0.5X MS medium supplemented with 20 μg/ml hygromycin. Resistant plants were transferred to soil. Flowers arising from the primary shoot were analysed by light microscopy. For the detailed analysis of the AP1::AP1, AP1::AG and AP1::AG[IAP1], the strong allele *ap1-7* was used as transformation background and T1 seeds were selected based on the seed specific GFP selection marker. Flower phenotypic analyses were performed by light microscopy on flower number ~10–19 based on their order of emergence on more than 20 T1 plants for each construct, growing in parallel to *ap1-7* plants. Three independent lines with one insertion expressing AG, AP1 and AG[IAP1] were selected for further phenotype analyses.

**Environmental SEM**. SEM experiments were performed at the Electron Microscopy Facility of the ICMG Nanobio-Chemistry Platform (Grenoble, France). Untreated flowers were directly placed in the microscope chamber. Care was taken to maintain humidity during the pressure decrease in the chamber in order to prevent tissue drying. Secondary electron images were recorded with a Quanta FEG 250 (FEI) microscope while maintaining the tissue at 2 °C, under a pressure of 500 Pa and a 70% relative humidity. The accelerating voltage was 14 kV and the image magnification ranged from 100 to 800×. Flowers from three independent lines were observed for each genotype.

## Data availability
Crystallographic data have been deposited with the PDB under the code 7NB0. SEP3-AG[AP1]I DAP-seq sequence data from this article can be found in the NCBI GEO data libraries under accession number GSE166205. *Arabidopsis* mutants are available upon request to the authors. Source data are provided with this paper.

## Code availability

Python, R and Shell scripts are available at https://github.com/Bioinfo-LPCV-RDF/TF_genomic_analysis_genomic_analysis and https://doi.org/10.5281/zenodo.5060328[76].

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

## Acknowledgements

This project received support from the ANR (ANR-16-CE92-0023) and GRAL, a programme from the Chemistry and Biology Health Graduate School of the University Grenoble Alpes (ANR-17-EURE-0003). This work benefited from access to the MX-Grenoble, an Instruct-ERIC centre within the Grenoble Partnership for Structural Biology. Financial support was provided by Instruct-ERIC (PID 13317) and FRISBI (ANR-10-INBS-05-02). R.V.-L. and C.S. were supported by the Deutsche Forschungsgemeinschaft (DFG project number 408478570).

## Author contributions

K.K., F.P., V.H., C.S. and C.Z. conceived the study, C.S., V.H., R.I., F.P. and C.Z. designed experiments, X.L., R.V.-L., C.S.S., A.J., M.H.N., F.v.d.W. and V.H. performed experiments, R.B.-M., J.L., M.H.N., J.M.M., F.P., C.S., K.K., R.I. and C.Z. analysed data and C.Z. wrote the manuscript with the help of all authors.

## Funding

## Competing interests

The authors declare no competing interests.
