## [Peer Review File · Nature Communications]

REVIEWERS' COMMENTS

Reviewer #4 (Remarks to the Author):

MADS review: June 15-22, 2021

The Manuscript NCOMMS-21-20994-T entitled "The Intervening Domain Is Required For DNA-binding and Functional Identity of Plant MADS Transcription Factors" by Drs. Lai ... et al. and Zubieta has clearly identified and confirmed in depth the I-domain (Intervening Domain) that has been defined previously in type II plant MADS transcription factors (TF) exists also in type I plant MADS, and the I-domain is important for its function as DNA-binding and protein-protein interactions (PPI) by a wide range of techniques including biochemical and structural (crystallography) methods; HTP sequencing Seq-DAP-seq, and ChIP-seq; and Yeast 2-hybrid experiments; particularly genetic domain-swapping of I-domains among different MADS TFs SEP3-AG and SEP3-AGIAP1 in planta. These work and results are of importance for deep understanding of the MADS TF family which is a conserved TF superfamily in eukaryotic organisms ranging from yeasts, plant and human. Thus, it deserves publication in a good journal such as NC.

Good words said as above, I do have some critics for some content and technique details:

In the abstract, the sentence "Here, we demonstrate that DNA binding in both lineages absolutely requires a short amino acid sequence C-terminal to the M domain called the Intervening domain (I domain) in type II MADS." May cause ambiguity, I suggest to change as "Here, we demonstrate that DNA binding in both lineages absolutely requires a short amino acid sequence C-terminal to the M domain called the Intervening domain (I domain) that was previously defined in type II MADS."

Some minor points:

- 1) The mutant protein expression and characterization may need (not absolutely required) to be checked by CD (circular dichroism) spectrometry to ensure their structural integrity;
- 2) Crystallographic data and structure determination: 2.1Å is a rather high resolution, but in the highest resolution shell (2.16-2.10) 130% is too high for Rmerge, please also list I/sig(I) for judgement of overall signal to noise ratio. Statistics will look better by cutting off the resolution a little, and will not lose any information. The structural determination and refinement have been described too simple or even omitted, please add anything necessary to validate the structural determination and analysis.
- 3) The structural labeling is also much too simplified, particularly for untrained eyes, please label names for helices and strands, particularly please label N- and C-termini. Please also label domain names, e.g. I-domain, where is it?

4) Without a DNA-TF complex structure, it's rather difficult to explain why the I-domain is so much important for DNA binding (and binding specificities), and PPIs, particularly as the authors claimed the I-domain is not even touching the DNA. Some vague arguments about allosteric change are rather weak in this case. Although this is a non-structure focused paper, some DNA-protein modeling including I-domain may be necessary. The following is some complex structural examples, mentioned by the authors in the paper.

(A PDF file containing the figures will be attached ...)

MADS review: June 15-22, 2021

The Manuscript NCOMMS-21-20994-T entitled "The Intervening Domain Is Required For DNA-binding and Functional Identity of Plant MADS Transcription Factors" by Drs. Lai ... et al. and Zubieta has clearly identified and confirmed in depth the I-domain (Intervening Domain) that has been defined previously in type II plant MADS transcription factors (TF) exists also in type I plant MADS, and the I-domain is important for its function as DNA-binding and protein-protein interactions (PPI) by a wide range of techniques including biochemical and structural (crystallography) methods; HTP sequencing Seq-DAP-seq, and ChIP-seq; and Yeast 2-hybrid experiments; particularly genetic domain-swapping of I-domains among different MADS TFs SEP3-AG and SEP3-AGIAP1 in planta. These work and results are of importance for deep understanding of the MADS TF family which is a conserved TF superfamily in eukaryotic organisms ranging from yeasts, plant and human. Thus, it deserves publication in a good journal such as NC.

Good words said as above, I do have some critics for some content and technique details:

In the abstract, the sentence "Here, we demonstrate that DNA binding in both lineages absolutely requires a short amino acid sequence C-terminal to the M domain called the Intervening domain (I domain) in type II MADS." May cause ambiguity, I suggest to change as "Here, we demonstrate that DNA binding in both lineages absolutely requires a short amino acid sequence C-terminal to the M domain called the Intervening domain (I domain) that was previously defined in type II MADS."

As suggested by the reviewer, we have made this change. We have also rewritten the abstract to be 165 words, as required.

Some minor points:

1) The mutant protein expression and characterization may need (not absolutely required) to be checked by CD (circular dichroism) spectrometry to ensure their structural integrity;

We used thermal shift assays to provide an overview of structural integrity and indeed it is highly variable depending on the mutation in the I domain. This is expected as the secondary structural elements exhibit many different interactions-hydrophobic, salt bridges, hydrogen bonding as described in the structure. CD would likely show a similar trend, although we are most interested in the quaternary structure which is efficiently monitored by thermal shift assay.

2) Crystallographic data and structure determination: 2.1A is a rather high resolution, but in the highest resolution shell (2.16-2.10) 130% is too high for Rmerge, please also list I/sig(I) for judgement of overall signal to noise ratio. Statistics will look better by cutting off the resolution a little, and will not lose any information. The structural determination and refinement have been described too simple or even omitted, please add anything necessary to validate the structural determination and analysis.

We used CC1/2 as the main criteria for determining resolution limits as this criterion is replacing Rmerge for high resolution limit determination. See, for example, Karplus and

Diedrichs, "Assessing and maximizing data quality in macromolecular crystallography," *Current Opinion in Structural Biology*. 2015 October; 34:60-68.
doi:10.1016/j.sbi.2015.07.003.

We have added details to the materials and methods section as to the data analysis, molecular replacement program and the refinement programs used for the structure building and analysis.

3) The structural labeling is also much too simplified, particularly for untrained eyes, please label names for helices and strands, particularly please label N- and C-termini. Please also label domain names, e.g. I-domain, where is it?

We have updated the Figure 1 accordingly with N and C termini labelled and the I domain circled in order to clarify the structural presentation. The figure legend has also been clarified as per the reviewer's suggestions.

4) Without a DNA-TF complex structure, it's rather difficult to explain why the I-domain is so much important for DNA binding (and binding specificities), and PPIs, particularly as the authors claimed the I-domain is not even touching the DNA. Some vague arguments about allosteric change are rather weak in this case. Although this is a non-structure focused paper, some DNA-protein modeling including I-domain may be necessary. The following is some complex structural examples, mentioned by the authors in the paper.

We agree that ideally, we would be able to present a structure bound to DNA, however after trying different length DNA, multiple blunt end and overhanging oligomers of varying length and different purifications of many protein-DNA complexes we were never able to obtain well-diffracting crystals of the complex. This difficulty in crystallization has been, unfortunately, noted for MADS TF-DNA complexes for many years (see, for example, Tan, et al. *J. Mol. Biol.*, 2000. 297, 947-959). However, we used structures of MEF2 and SRF that were bound to DNA for our structural comparisons (see Figure 1). The M domain overlays of MEF2 (human), SRF (yeast) and SEP3 (plant, this study) demonstrate the conservation of this domain in all eukaryotes and, as the structures are with DNA (MEF2 and SRF) and without DNA (SEP3, this study), the low rmsd for the structural alignments shows that there is little structural change in the protein upon DNA binding. The amino acid conservation of the M domain in different kingdoms of life, particularly amino acids that directly contact the DNA, coupled with functional studies demonstrating that M domains from human and yeast will complement in *Arabidopsis* all suggest that the M domain is not sufficient for DNA-binding specificity. In order to reconcile these data, we examined the I domain and show that it is required for DNA-binding specificity. Based on all available data this is not due to direct contacts with the DNA but rather allosteric effects and altered protein partner recruitment.